# Direct observation of orbital hybridisation in a cuprate superconductor

C.E. Matt [1,2], D. Sutter [1], A.M. Cook[1], Y. Sassa[3], M. Månsson [4], O. Tjernberg [4], L. Das[1], M. Horio [1], D. Destraz[1], C.G. Fatuzzo[5], K. Hauser[1], M. Shi[2], M. Kobayashi[2], V.N. Strocov[2], T. Schmitt[2], P. Dudin[6], M. Hoesch[6], S. Pyon[7], T. Takayama[7], H. Takagi [7], O.J. Lipscombe[8], S.M. Hayden[8], T. Kurosawa[9], N. Momono[9,10], M. Oda[9], T. Neupert[1] & J. Chang [1]

The minimal ingredients to explain the essential physics of layered copper-oxide (cuprates) materials remains heavily debated. Effective low-energy single-band models of the copper–oxygen orbitals are widely used because there exists no strong experimental evidence supporting multi-band structures. Here, we report angle-resolved photoelectron spectroscopy experiments on La-based cuprates that provide direct observation of a two-band structure. This electronic structure, qualitatively consistent with density functional theory, is parametrised by a two-orbital ($d_{x^2-y^2}$ and $d_{z^2}$) tight-binding model. We quantify the orbital hybridisation which provides an explanation for the Fermi surface topology and the proximity of the van-Hove singularity to the Fermi level. Our analysis leads to a unification of electronic hopping parameters for single-layer cuprates and we conclude that hybridisation, restraining $d$-wave pairing, is an important optimisation element for superconductivity.

[1] Physik-Institut, Universität Zürich, Winterthurerstrasse 190, CH-8057 Zürich, Switzerland. [2] Swiss Light Source, Paul Scherrer Institut, CH-5232 Villigen PSI, Switzerland. [3] Department of Physics and Astronomy, Uppsala University, SE-75121 Uppsala, Sweden. [4] Materials Physics, KTH Royal Institute of Technology, SE-164 40 Kista, Stockholm, Sweden. [5] Institute of Physics, École Polytechnique Fedérale de Lausanne (EPFL), Lausanne CH-1015, Switzerland. [6] Diamond Light Source, Harwell Campus, Didcot OX11 0DE, UK. [7] Department of Advanced Materials, University of Tokyo, Kashiwa 277-8561, Japan. [8] H. H. Wills Physics Laboratory, University of Bristol, Bristol BS8 1TL, UK. [9] Department of Physics, Hokkaido University, Sapporo 060-0810, Japan. [10] Department of Applied Sciences, Muroran Institute of Technology, Muroran 050-8585, Japan. Correspondence and requests for materials should be addressed to C.E.M. (email: cmatt@g.harvard.edu) or to J.C. (email: johan.chang@physik.uzh.ch)

I dentifying the factors that limit the transition temperature $T_c$ of high-temperature cuprate superconductivity is a crucial step towards revealing the design principles underlying the pairing mechanism[1]. It may also provide an explanation for the dramatic variation of $T_c$ across the known single-layer compounds[2]. Although superconductivity is certainly promoted within the copper-oxide layers, the apical oxygen position may play an important role in defining the transition temperature[3–7]. The $CuO_6$ octahedron lifts the degeneracy of the nine copper $3d$-electrons and generates fully occupied $t_{2g}$ and 3/4-filled $e_g$ states[8]. With increasing apical oxygen distance $d_A$ to the $CuO_2$ plane, the $e_g$ states split to create a 1/2-filled $d_{x^2-y^2}$ band. The distance $d_A$ thus defines whether single or two-band models are most appropriate to describe the low-energy band structure. It has also been predicted that $d_A$ influences $T_c$ in at least two different ways. First, the distance $d_A$ controls the charge transfer gap between the oxygen and copper site which, in turn, suppresses superconductivity[5,9]. Second, Fermi-level $d_{z^2}$ hybridisation, depending on $d_A$, reduces the pairing strength[6,10]. Experimental evidence, however, points in opposite directions. Generally, single-layer materials with larger $d_A$ have indeed a larger $T_c$[2]. However, scanning tunneling microscopy (STM) studies of Bi-based cuprates suggest an anti-correlation between $d_A$ and $T_c$[11].

In the quest to disentangle these causal relation between $d_A$ and $T_c$, it is imperative to experimentally reveal the orbital character of the cuprate band structure. The comparably short apical oxygen distance $d_A$ makes $La_{2-x}Sr_xCuO_4$ (LSCO) an ideal candidate for such a study. Experimentally, however, it is challenging to determine the orbital character of the states near the Fermi energy ($E_F$). In fact, the $d_{z^2}$ band has never been identified directly by angle-resolved photoelectron spectroscopy (ARPES) experiments. A large majority of ARPES studies have focused on the pseudogap, superconducting gap and quasi-particle self-energy properties in near vicinity to the Fermi level[12]. An exception to this trend are studies of the so-called

waterfall structure[13–17] that lead to the observation of band structures below the $d_{x^2-y^2}$ band[14,16]. However, the origin and hence orbital character of these bands was never addressed. Resonant inelastic X-ray scattering has been used to probe excitations between orbital $d$-levels. In this fashion, insight about the position of $d_{z^2}$, $d_{xz}$, $d_{yz}$ and $d_{xy}$ states with respect to $d_{x^2-y^2}$ has been obtained[18]. Although difficult to disentangle, it has been argued that for LSCO the $d_{z^2}$ level is found above $d_{xz}$, $d_{yz}$ and $d_{xy}$[19,20]. To date, a comprehensive study of the $d_{z^2}$ momentum dependence is missing and therefore the coupling between the $d_{z^2}$ and $d_{x^2-y^2}$ bands has not been revealed. X-ray absorption spectroscopy (XAS) experiments, sensitive to the unoccupied states, concluded only marginal hybridisation of $d_{x^2-y^2}$ and $d_{z^2}$ states in LSCO[21]. Therefore, the role of $d_{z^2}$ hybridisation remains ambiguous[22].

Here we provide direct ultraviolet and soft-X-ray ARPES measurements of the $d_{z^2}$ band in La-based single-layer compounds. The $d_{z^2}$ band is located about 1 eV below the Fermi level at the Brillouin zone (BZ) corners. From these corners, the $d_{z^2}$ band is dispersing downwards along the nodal and anti-nodal directions, consistent with density functional theory (DFT) calculations. The experimental and DFT band structure, including only $d_{x^2-y^2}$ and $d_{z^2}$ orbitals, is parametrised using a two-orbital tight-binding model[23]. The presence of the $d_{z^2}$ band close to the Fermi level allows to describe the Fermi surface topology for all single-layer compounds (including $HgBa_2CuO_{4+x}$ and $Tl_2Ba_2$-$CuO_{6+x}$) with similar hopping parameters for the $d_{x^2-y^2}$ orbital. This unification of electronic parameters implies that the main difference between single-layer cuprates originates from the hybridisation between $d_{x^2-y^2}$ and $d_{z^2}$ orbitals. The significantly increased hybridisation in La-based cuprates pushes the van-Hove singularity close to the Fermi level. This explains why the Fermi surface differs from other single-layer compounds. We directly quantify the orbital hybridisation that plays a sabotaging role for superconductivity.

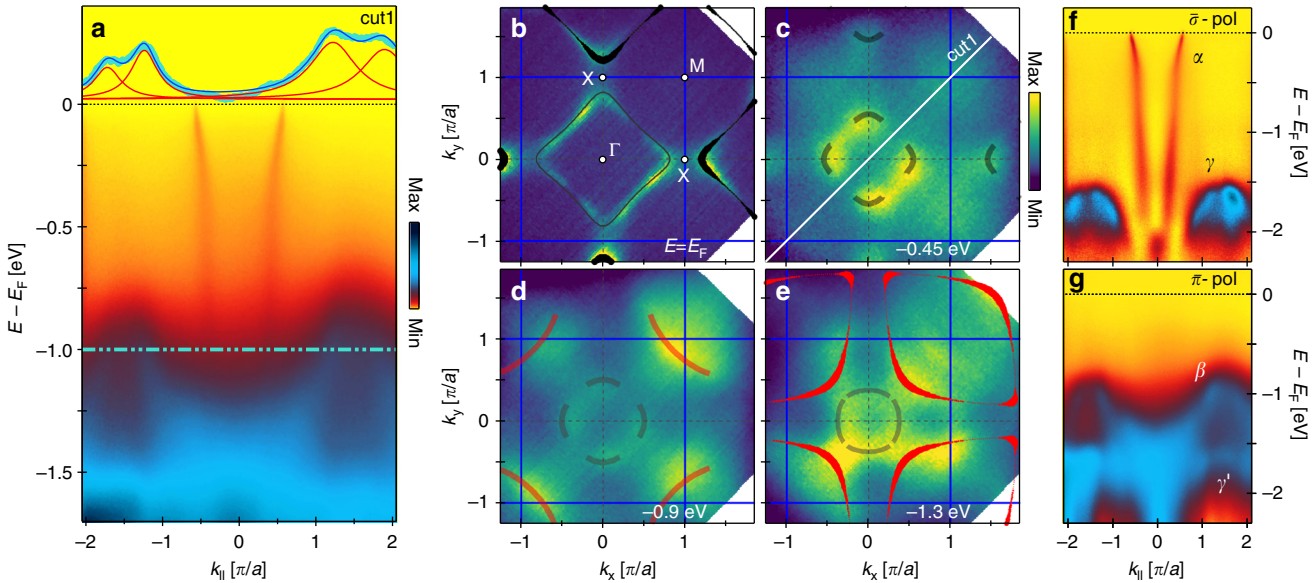

**Fig. 1** ARPES spectra showing $e_g$-bands of overdoped $La_{2-x}Sr_xCuO_4$ $x = 0.23$. **a** Raw ARPES energy distribution map (EDM) along cut 1 as indicated in (**c**). Dashed green line indicates the position of MDC displayed on top by turquoise circles. A linear background has been subtracted from the MDC which is fitted (blue line) by four Lorentzians (red lines). **b–e** Constant binding energy maps at $E_F$ (**b**) and at higher binding energies (**c–e**) as indicated. The photoemission intensity, shown in false colour scale, is integrated over ± 10 meV. Black (red) lines indicate the position of $d_{x^2-y^2}$ ($d_{z^2}$) bands. The curve thickness in **b**, **e** is scaled to the contribution of the $d_{z^2}$ orbital. Semitransparent lines are guides to the eye. **f**, **g** EDMs along cut 1 recorded with $\bar{\sigma}$ and $\bar{\pi}$ light, **f** sensitive to the low-energy $d_{x^2-y^2}$ and $d_{xz}/d_{yz}$ bands and **g** the $d_{z^2}$ and $d_{xy}$-derived bands. All data have been recorded with $h\nu = 160$ eV

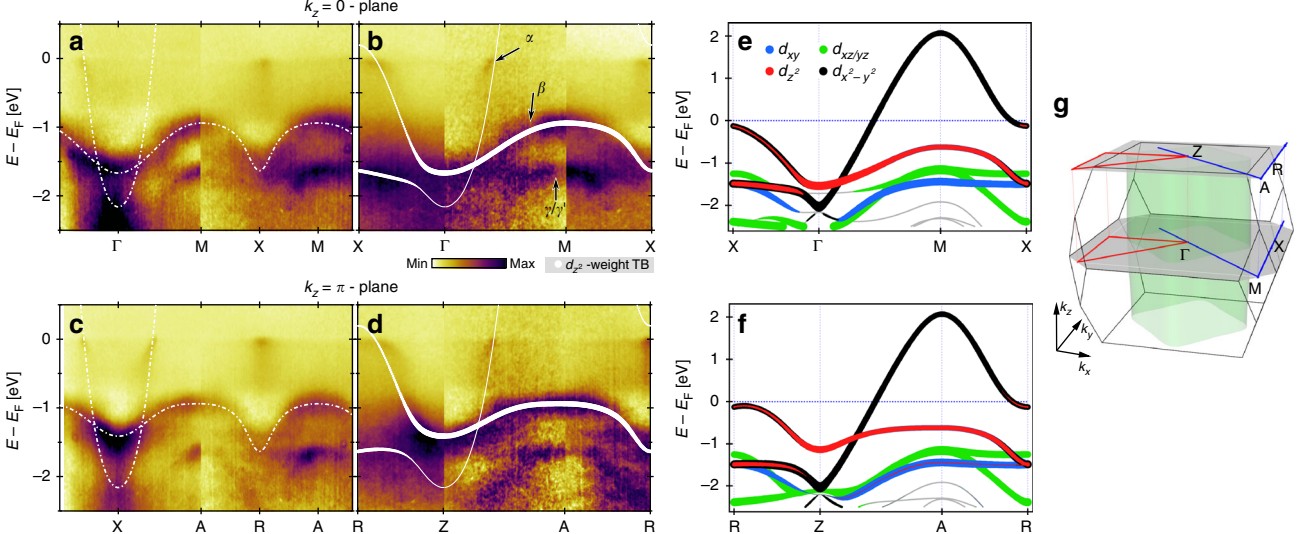

**Fig. 2** Comparison of observed and calculated band structure. **a–d** Background subtracted (see Methods section) soft-X-ray ARPES EDMs recorded on La$_{2-x}$Sr$_x$CuO$_4$, $x = 0.23$ along in-plane high-symmetry directions for $k_z = 0$ and $k_z = \pi/c'$ as indicated in **g**. White lines represent the two-orbital ($d_{z^2}$ and $d_{x^2-y^2}$) tight-binding model as described in the text. The line width in **b**, **d** indicates the orbital weight of the $d_{z^2}$ orbital. **e, f** Corresponding in-plane DFT band structure at $k_z = 0$ and $k_z = \pi/c'$, calculated for La$_2$CuO$_4$ (see Methods section). The colour code indicates the orbital character of the bands. Around the anti-nodal points (X or R), strong hybridisation of $d_{z^2}$ and $d_{x^2-y^2}$ orbitals is found. In contrast, symmetry prevents any hybridisation along the nodal lines (Γ–M or Z–A). **g** Sketch of the 3D BZ of LSCO with high symmetry lines and points as indicated

## Results

**Material choices**. Different dopings of LSCO spanning from $x = 0.12$ to 0.23 in addition to an overdoped compound of La$_{1.8-x}$Eu$_{0.2}$Sr$_x$CuO$_4$ with $x = 0.21$ have been studied. These compounds represent different crystal structures: low-temperature orthorhombic, low-temperature tetragonal and the high-temperature tetragonal. Our results are very similar across all crystal structures and dopings (Supplementary Fig. 1). To keep the comparison to band structure calculations simple, this paper focuses on results obtained in the tetragonal phase of overdoped LSCO with $x = 0.23$.

**Electronic band structure**. A raw ARPES energy distribution map (EDM), along the nodal direction, is displayed in Fig. 1a. Near $E_F$, the widely studied nodal quasiparticle dispersion with predominately $d_{x^2-y^2}$ character is observed[12]. This band reveals the previously reported electron-like Fermi surface of LSCO, $x = 0.23$[24,25] (Fig. 1b), the universal nodal Fermi velocity $v_F \approx 1.5$ eVÅ[26] and a band dispersion kink around 70 meV[26]. The main observation reported here is the second band dispersion at ~1 eV below the Fermi level $E_F$ (Figs. 1 and 2) and a hybridisation gap splitting the two (Fig. 3). This second band—visible in both raw momentum distribution curves (MDC) and constant energy maps—disperses downwards away from the BZ corners. Since a pronounced $k_z$ dependence is observed for this band structure (Figs. 2 and 4) a trivial surface state can be excluded. Subtracting a background intensity profile (Supplementary Fig. 2) is a standard method that enhances visualisation of this second band structure. In fact, using soft X-rays (160–600 eV), at least two additional bands ($\beta$ and $\gamma$) are found below the $d_{x^2-y^2}$ dominated band crossing the Fermi level. Here, focus is set entirely on the $\beta$ band dispersion closest to the $d_{x^2-y^2}$ dominated band. This band is clearly observed at the BZ corners (Figs. 1–3). The complete in-plane ($k_x$, $k_y$) and out-of-plane ($k_z$) band dispersion is presented in Fig. 4.

**Orbital band characters**. To gain insight into the orbital character of these bands, a comparison with a DFT band structure

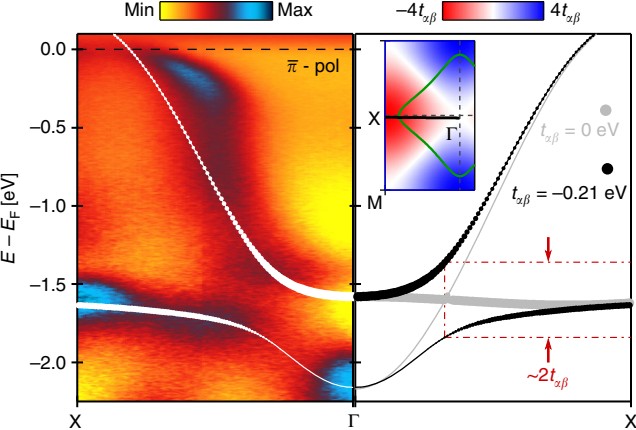

**Fig. 3** Avoided band crossing. Left panel: ultraviolet ARPES data recorded along the ant-inodal direction using 160 eV linear horizontal polarised photons. Solid white lines are the same tight-binding model as shown in Fig. 2. Right panel: tight-binding model of the $d_{x^2-y^2}$ and $d_{z^2}$ bands along the anti-nodal direction. Grey lines are the model prediction in absence of inter-orbital hopping ($t_{\alpha\beta} = 0$) between $d_{x^2-y^2}$ and $d_{z^2}$. In this case, the bands are crossing near the Γ-point. This degeneracy is lifted once a finite inter-orbital hopping parameter is considered. For solid black lines $t_{\alpha\beta} = -210$ meV and other hopping parameters have been adjusted accordingly. Inset indicates the Fermi surface (green line) and the Γ − X cut directions. Coloured background displays the amplitude of the hybridisation term Ψ(**k**) that vanishes on the nodal lines

calculation (see Methods section) of La$_2$CuO$_4$ is shown in Fig. 2. The $e_g$ states ($d_{x^2-y^2}$ and $d_{z^2}$) are generally found above the $t_{2g}$ bands ($d_{xy}$, $d_{xz}$ and $d_{yz}$). The overall agreement between the experiment and the DFT calculation (Supplementary Fig. 3) thus suggests that the two bands nearest to the Fermi level are composed predominantly of $d_{x^2-y^2}$ and $d_{z^2}$ orbitals. This conclusion can also be reached by pure experimental arguments. Photo-emission matrix element selection rules contain information

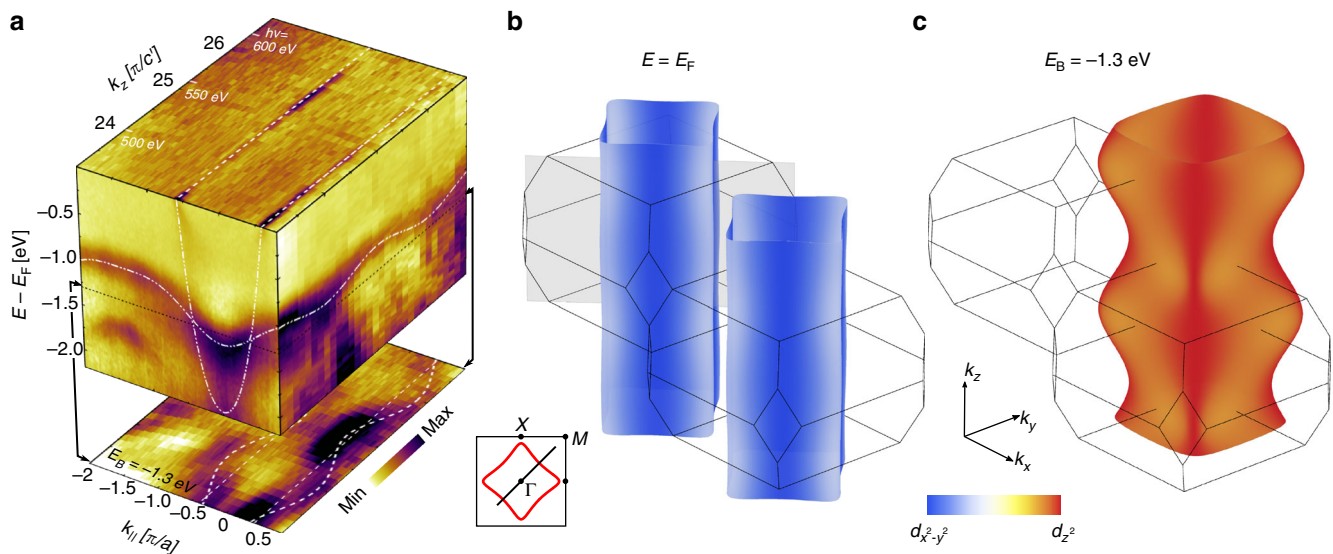

**Fig. 4** Three-dimensional band dispersion. **a** $k_z$ dispersion recorded along the diagonal $(\pi, \pi)$ direction of the $d_{x^2-y^2}$ and $d_{z^2}$ bands (along grey plane in **b**). Whereas the $d_{x^2-y^2}$ band displays no $k_z$ dependence beyond matrix element effects, the $d_{z^2}$ band displays a discernible $k_z$ dispersion. The iso-energy map below the cube has binding energy $E - E_F = -1.3$ eV. White lines represent the tight-binding model. **b**, **c** Tight-binding representation of the Fermi surface ($\alpha$ band) and iso-energy surface ($-1.3$ eV) of the $\beta$ band. The colour code indicates the **k**-dependent orbital hybridisation. The orbital hybridisation at $E_F$ is largest in the anti-nodal region of the $k_z = \pi/c'$ plane where the $d_{z^2}$ admixture at $k_F$ amounts to ~1/3

**Table 1 Tight-binding parameters for single-layer cuprate materials**

| Compound | LSCO | Hg1201 | Tl2201 | LSCO |
|---|---|---|---|---|
| Doping $p$ | 0.22 | 0.16 | 0.26 | 0.23 |
| Tight binding parameters in units of $t_\alpha = -1.21$ eV | | | | |
| $-\mu$ | 0.88 | 1.27 | 1.35 | 0.96 |
| $-t'_\alpha$ | 0.13 | 0.47 | 0.42 | 0.32 |
| $t''_\alpha$ | 0.065 | 0.02 | 0.02 | 0.0 |
| $t_{\alpha\beta}$ | 0 | 0 | 0 | 0.175 |
| $t_\beta$ | – | – | – | 0.062 |
| $t'_\beta$ | – | – | – | 0.017 |
| $t_{\beta z}$ | – | – | – | 0.017 |
| $-t'_{\beta z}$ | – | – | – | 0.0017 |
| Ref. | 24 | 39,40 | 41,42 | This work |

Comparison of tight-binding hopping parameters obtained from single-orbital and two-orbital models. Once a coupling $t_{\alpha\beta}$ between the $d_{x^2-y^2}$ and $d_{z^2}$ band is introduced for La$_{2-x}$Sr$_x$CuO$_4$, the $d_{x^2-y^2}$ hopping parameters become comparable to those of Hg1201 and Tl2201

about the orbital band character. They can be probed in a particular experimental setup where a mirror-plane is defined by the incident light and the electron analyser slit[12]. With respect to this plane the electromagnetic light field has odd (even) parity for $\overline{\sigma}$ ($\overline{\pi}$) polarisation (Supplementary Fig. 4). Orienting the mirror plane along the nodal direction (cut 1 in Fig. 1), the $d_{z^2}$ and $d_{xy}$ ($d_{x^2-y^2}$) orbitals have even (odd) parity. For a final-state with even parity, selection rules[12] dictate that the $d_{z^2}$ and $d_{xy}$-derived bands should appear (vanish) in the $\overline{\pi}$ ($\overline{\sigma}$) polarisation channel and vice versa for $d_{x^2-y^2}$. Due to their orientation in real-space, the $d_{xz}$ and $d_{yz}$ orbitals are not expected to show a strict switching behaviour along the nodal direction[27]. As shown in Fig. 1f, g, two bands ($\alpha$ and $\gamma$) appear with $\overline{\sigma}$-polarised light while for $\overline{\pi}$-polarised light bands $\beta$ and $\gamma'$ are observed. Band $\alpha$ which crosses $E_F$ is assigned to $d_{x^2-y^2}$ while band $\gamma$ has to originate from $d_{xz}/d_{yz}$ orbitals as $d_{z^2}$ and $d_{xy}$-derived states are fully suppressed for $\overline{\sigma}$-polarised light. In the EDM, recorded with $\overline{\pi}$-polarised light, band ($\beta$) at ~1 eV binding energy and again a band ($\gamma'$) at ~1.6 eV

is observed. From the orbital shape, a smaller $k_z$ dispersion is expected for $d_{x^2-y^2}$ and $d_{xy}$-derived bands than for those from $d_{z^2}$ orbitals. As the $\beta$ band exhibits a significant $k_z$ dispersion (Fig. 4), much larger than observed for the $d_{x^2-y^2}$ band, we conclude that it is of $d_{z^2}$ character. The $\gamma'$ band which is very close to the $\gamma$ band is therefore of $d_{xy}$ character. Interestingly, this $d_{z^2}$-derived band has stronger in-plane than out-of-plane dispersion, suggesting that there is a significant hopping to in-plane $p_x$ and $p_y$ oxygen orbitals. Therefore the assumption that the $d_{z^2}$ states are probed uniquely through the apical oxygen $p_z$ orbital[21] has to be taken with caution.

## Discussion

Most minimal models aiming to describe the cuprate physics start with an approximately half-filled single $d_{x^2-y^2}$ band on a two-dimensional square lattice. Experimentally, different band structures have been observed across single-layer cuprate compounds. The Fermi surface topology of LSCO is, for example, less rounded compared to (Bi,Pb)$_2$(Sr,La)$_2$CuO$_{6+x}$ (Bi2201), Tl$_2$Ba$_2$CuO$_{6+x}$ (Tl2201) and HgBa$_2$CuO$_{4+x}$ (Hg1201). Within a single-band tight-binding model the rounded Fermi surface shape of the single-layer compounds Hg1201 and Tl2201 is described by setting $r = (|t'_\alpha| + |t''_\alpha|)/t_\alpha \sim 0.4$[6], where $t_\alpha$, $t'_\alpha$ and $t''_\alpha$ are nearest neighbour (NN), next–nearest neighbour (NNN) and next-next–nearest neighbour (NNNN) hopping parameters (Table 1 and Supplementary Fig. 4). For LSCO with more flat Fermi surface sections, significantly lower values of $r$ have been reported. For example, for overdoped La$_{1.78}$Sr$_{0.22}$CuO$_4$, $r \sim 0.2$ was found[24,25]. The single-band premise thus leads to varying hopping parameters across the cuprate families, stimulating the empirical observation that $T_c^{max}$ roughly scales with $t'^2_\alpha$. This, however, is in direct contrast to $t$–$J$ models that predict the opposite correlation[28,29]. Thus the single-band structure applied broadly to all single-layer cuprates lead to conclusions that challenge conventional theoretical approaches.

The observation of the $d_{z^2}$ band calls for a re-evaluation of the electronic structure in La-based cuprates using a two-orbital

tight-binding model (see Methods section). Crucially, there is a hybridisation term $\Psi(\mathbf{k}) = 2t_{\alpha\beta}\left[\cos(k_x a) - \cos(k_y b)\right]$ between the $d_{x^2-y^2}$ and $d_{z^2}$ orbitals, where $t_{\alpha\beta}$ is a hopping parameter that characterises the strength of orbital hybridisation. In principle, one may attempt to describe the two observed bands independently by taking $t_{\alpha\beta} = 0$. However, the problem then returns to the single-band description with the above mentioned contradictions. Furthermore, $t_{\alpha\beta} = 0$ implies a band crossing in the anti-nodal direction that is not observed experimentally (Fig. 3). In fact, from the avoided band crossing one can directly estimate $t_{\alpha\beta} \approx -200$ meV. As dictated by the different eigenvalues of the orbitals under mirror symmetry, the hybridisation term $\Psi(\mathbf{k})$ vanishes on the nodal lines $k_x = \pm k_y$ (see inset of Fig. 3). Hence the pure $d_{x^2-y^2}$ and $d_{z^2}$ orbital band character is expected along these nodal lines. The hybridisation $\Psi(\mathbf{k})$ is largest in the anti-nodal region, pushing the van-Hove singularity of the upper band close to the Fermi energy and in case of overdoped LSCO across the Fermi level.

In addition to the hybridisation parameter $t_{\alpha\beta}$ and the chemical potential $\mu$, six free parameters enter the tight-binding model that yields the entire band structure (white lines in Figs. 2 and 4). Nearest and next-nearest in-plane hopping parameters between $d_{x^2-y^2}$ ($t_\alpha$, $t'_\alpha$) and $d_{z^2}$ ($t_\beta$, $t'_\beta$) orbitals are introduced to capture the Fermi surface topology and in-plane $d_{z^2}$ band dispersion (Supplementary Fig. 4). The $k_z$ dispersion is described by nearest and next-nearest out-of-plane hoppings ($t_{\beta z}$, $t'_{\beta z}$) of the $d_{z^2}$ orbital. The four $d_{z^2}$ hopping parameters and the chemical potential $\mu$ are determined from the experimental band structure along the nodal direction where $\Psi(\mathbf{k}) = 0$. Furthermore, the $\alpha$ and $\beta$ band dispersion in the anti-nodal region and the Fermi surface topology provide the parameters $t_\alpha$, $t'_\alpha$ and $t_{\alpha\beta}$. Our analysis reveals a finite band coupling $t_{\alpha\beta} = -0.21$ eV resulting in a strong anti-nodal orbital hybridisation (Fig. 2 and Table 1). Compared to the single-band parametrisation[24] a significantly larger value $r \sim -0.32$ is found and hence a unification of $t'_\alpha/t_\alpha$ ratios for all single-layer compounds is achieved.

Finally, we discuss the implication of orbital hybridisation for superconductivity and pseudogap physics. First, we notice that a pronounced pseudogap is found in the anti-nodal region of $\mathrm{La_{1.8-x}Eu_{0.2}Sr_xCuO_4}$ with $x = 0.21$—consistent with transport experiments[30] (Supplementary Fig. 5). The fact that $t_{\alpha\beta}$ of $\mathrm{La_{1.59}Eu_{0.2}Sr_{0.21}CuO_4}$ is similar to $t_{\alpha\beta}$ of LSCO suggests that the pseudogap is not suppressed by the $d_{z^2}$ hybridisation. To this end, a comparison to the 1/4-filled $e_g$ system $\mathrm{Eu_{2-x}Sr_xNiO_4}$ with $x = 1.1$ is interesting[31,32]. This material has the same two-orbital band structure with protection against hybridisation along the nodal lines. Both the $d_{x^2-y^2}$ and $d_{z^2}$ bands are crossing the Fermi level, producing two Fermi surface sheets[31]. Despite an even stronger $d_{z^2}$ admixture of the $d_{x^2-y^2}$ derived band a $d$-wave-like pseudogap has been reported[32]. The pseudogap physics thus seems to be unaffected by the orbital hybridisation.

It has been argued that orbital hybridisation—of the kind reported here—is unfavourable for superconducting pairing[6,10]. It thus provides an explanation for the varying $T_c^{\max}$ across single-layer cuprate materials. Although other mechanisms, controlled by the apical oxygen distance, (e.g. variation of the copper–oxygen charge transfer gap[4]) are not excluded our results demonstrate that orbital hybridisation exists and is an important control parameter for superconductivity.

## Methods

**Sample characterisation.** High-quality single crystals of LSCO, $x = 0.12$, 0.23, and $\mathrm{La_{1.8-x}Eu_{0.2}Sr_xCuO_4}$, $x = 0.21$, were grown by the floating-zone technique. The samples were characterised by SQUID magnetisation[33] to determine superconducting transition temperatures ($T_c = 27$, 24 and 14 K). For the crystal structure, the experimental lattice parameters are $a = b = 3.78$ Å and $c = 2c' = 13.2$ Å[34].

**ARPES experiments.** Ultraviolet and soft-X-ray ARPES experiments were carried out at the SIS[43] and ADRESS[44] beam-lines at the Swiss Light Source and at the I05 beamline at Diamond Light Source. Samples were pre-aligned ex situ using a X-ray LAUE instrument and cleaved in situ—at base temperature (10–20 K) and ultra high vacuum ($\leq 5 \times 10^{-11}$ mbar)—employing a top-post technique or cleaving device[35]. Ultraviolet (soft X-ray[36]) ARPES spectra were recorded using a SCIENTA R4000 (SPECS PHOIBOS-150) electron analyser with horizontal (vertical) slit setting. All data was recorded at the cleaving temperature 10–20 K. To visualise the $d_{z^2}$-dominated band, we subtracted in Fig. 1f, g and Figs. 2–4 the background that was obtained by taking the minimum intensity of the MDC at each binding energy.

**Tight-binding model.** A two-orbital tight-binding model Hamiltonian with symmetry-allowed hopping terms is employed to isolate and characterise the extent of orbital hybridisation of the observed band structure[23]. For compactness of the momentum-space Hamiltonian matrix representation, we introduce the vectors

$$
\begin{aligned}
\mathbf{Q}^\kappa &= (a, \kappa b, 0)^\top, \\
\mathbf{R}^{\kappa_1,\kappa_2} &= (\kappa_1 a, \kappa_1\kappa_2 b, c)^\top/2, \\
\mathbf{T}_1^{\kappa_1,\kappa_2} &= (3\kappa_1 a, \kappa_1\kappa_2 b, c)^\top/2, \\
\mathbf{T}_2^{\kappa_1,\kappa_2} &= (\kappa_1 a, 3\kappa_1\kappa_2 b, c)^\top/2,
\end{aligned}
\tag{1}
$$

where $\kappa$, $\kappa_1$ and $\kappa_2$ take values $\pm 1$ as defined by sums in the Hamiltonian and $\top$ denotes vector transposition.

Neglecting the electron spin (spin–orbit coupling is not considered) the momentum-space tight-binding Hamiltonian, $\mathcal{H}(\mathbf{k})$, at a particular momentum $\mathbf{k} = (k_x, k_y, k_z)$ is then given by

$$
\mathcal{H}(\mathbf{k}) = \begin{bmatrix} M^{x^2-y^2}(\mathbf{k}) & \Psi(\mathbf{k}) \\ \Psi(\mathbf{k}) & M^{z^2}(\mathbf{k}) \end{bmatrix},
\tag{2}
$$

in the basis $\left(c_{\mathbf{k},x^2-y^2}, c_{\mathbf{k},z^2}\right)^\top$, where the operator $c_{\mathbf{k},\alpha}$ annihilates an electron with momentum $\mathbf{k}$ in an $e_g$-orbital $d_\alpha$, with $\alpha \in \{x^2 - y^2, z^2\}$. The diagonal matrix entries are given by

$$
\begin{aligned}
M^{x^2-y^2}(\mathbf{k}) = {} & 2t_\alpha\left[\cos(k_x a) + \cos(k_y b)\right] + \mu \\
& + \sum_{\kappa=\pm 1} 2t'_\alpha\cos(\mathbf{Q}^\kappa \cdot \mathbf{k}) \\
& + 2t''_\alpha\left[\cos(2k_x a) + \cos(2k_y b)\right],
\end{aligned}
\tag{3}
$$

and

$$
\begin{aligned}
M^{z^2}(\mathbf{k}) = {} & 2t_\beta\left[\cos(k_x a) + \cos(k_y b)\right] - \mu \\
& + \sum_{\kappa=\pm 1} 2t'_\beta\cos(\mathbf{Q}^\kappa \cdot \mathbf{k}) \\
& + \sum_{\kappa_1,2=\pm 1}\left[2t_{\beta z}\cos(\mathbf{R}^{\kappa_1,\kappa_2} \cdot \mathbf{k})\right. \\
& \left. + \sum_{i=1,2} 2t'_{\beta z}\cos(\mathbf{T}_i^{\kappa_1,\kappa_2} \cdot \mathbf{k})\right],
\end{aligned}
\tag{4}
$$

which describe the intra-orbital hopping for $d_{x^2-y^2}$ and $d_{z^2}$ orbitals, respectively. The inter-orbital nearest-neighbour hopping term is given by

$$
\Psi(\mathbf{k}) = 2t_{\alpha\beta}\left[\cos(k_x a) - \cos(k_y b)\right].
\tag{5}
$$

In the above, $\mu$ determines the chemical potential. The hopping parameters $t_\alpha$, $t'_\alpha$ and $t''_\alpha$ characterise NN, NNN and NNNN intra-orbital in-plane hopping between $d_{x^2-y^2}$ orbitals. $t_\beta$ and $t'_\beta$ characterise NN and NNN intra-orbital in-plane hopping between $d_{z^2}$ orbitals, while $t_{\beta z}$ and $t_{\beta z}$ characterise NN and NNN intra-orbital out-of-plane hopping between $d_{z^2}$ orbitals, respectively (Supplementary Fig. 3). Finally, the hopping parameter $t_{\alpha\beta}$ characterises NN inter-orbital in-plane hopping. Note that in our model, $d_{x^2-y^2}$ intraorbital hopping terms described by the vectors (Eq. (1)) are neglected as these are expected to be weak compared to those of the $d_{z^2}$ orbital. This is due to the fact that the inter-plane hopping is mostly mediated by hopping between apical oxygen $p_z$ orbitals, which in turn only hybridise with the $d_{z^2}$ orbitals, not with the $d_{x^2-y^2}$ orbitals. Such an argument highlights that the tight-binding model is not written in atomic orbital degrees of freedom, but in Wannier orbitals, which are formed from the Cu $d$ orbitals and the ligand oxygen $p$ orbitals. As follows from symmetry considerations and is discussed in ref. [10], the Cu $d_{z^2}$ orbital together with the apical oxygen $p_z$ orbital forms a Wannier orbital with $d_{z^2}$ symmetry, while the Cu $d_{x^2-y^2}$ orbital together with the four neighbouring $p_\sigma$ orbitals of the in-plane oxygen forms a Wannier orbital with $d_{x^2-y^2}$ symmetry. One should thus think of this tight-binding model as written in terms of these Wannier orbitals, thus implicitly containing superexchange hopping via the ligand oxygen $p$ orbitals. Additionally we stress that all hopping parameters effectively include the oxygen orbitals. Diagonalising Hamiltonian (2), we find two

bands

$$\varepsilon_{\pm}(\mathbf{k}) = \frac{1}{2}\left[M^{x^2-y^2}(\mathbf{k}) + M^{z^2}(\mathbf{k})\right]$$
$$\pm \frac{1}{2}\sqrt{\left[M^{x^2-y^2}(\mathbf{k}) - M^{z^2}(\mathbf{k})\right]^2 + 4\Psi^2(\mathbf{k})}, \qquad (6)$$

and make the following observations: along the $k_x = \pm k_y$ lines, $\Psi(\mathbf{k})$ vanishes and hence no orbital mixing appears in the nodal directions. The reason for this absence of mixing lies in the different mirror eigenvalues of the two orbitals involved. Hence it is not an artifact of the finite range of hopping processes included in our model. The parameters of the tight-binding model are determined by fitting the experimental band structure and are provided in Table 1.

**DFT calculations**. DFT calculations were performed for $La_2CuO_4$ in the tetragonal space group $I4/mmm$, No. 139, found in the overdoped regime of LSCO using the WIEN2K package[37]. Atomic positions are those inferred from neutron diffraction measurements[34] for $x = 0.225$. In the calculation, the Kohn–Sham equation is solved self-consistently by using a full-potential linear augmented plane wave (LAPW) method. The self consistent field calculation converged properly for a uniform k-space grid in the irreducible BZ. The exchange-correlation term is treated within the generalised gradient approximation in the parametrisation of Perdew, Burke and Enzerhof[38]. The plane wave cutoff condition was set to $RK_{max} = 7$ where $R$ is the radius of the smallest LAPW sphere (i.e. 1.63 times the Bohr radius) and $K_{max}$ denotes the plane wave cutoff.

**Data availability**. All experimental data are available upon request to the corresponding authors.

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

## Acknowledgements

D.S., D.D., L.D., T.N., C.E.M, C.G.F. and J.C. acknowledge support by the Swiss National Science Foundation. Further, Y.S. and M.M. are supported by the Swedish Research Council (VR) through a project (BIFROST, dnr.2016-06955). O.T. acknowledges support from the Swedish Research Council as well as the Knut and Alice Wallenberg foundation. This work was performed at the SIS, ADRESS and I05 beamlines at the Swiss Light Source and at the Diamond Light Source. A.M.C. wishes to thank the Aspen Center for Physics, which is supported by National Science Foundation grant PHY-1066293, for hosting during some stages of this work. We acknowledge Diamond Light Source for access to beamline I05 (proposal SI10550-1) that contributed to the results presented here and thank all the beamline staff for technical support.

## Author contributions

S.P., T.T., H.T., T.K., N.M., M.O., O.J.L. and S.M.H. grew and prepared single crystals. C.E.M., D.S., L.D., M.H., D.D., C.G.F., K.H., J.C., M.S., O.T., M.K., V.N.S., T.S., P.D., M.H., M.M. and Y.S. prepared and carried out the ARPES experiment. C.E.M., K.H. and J.C. performed the data analysis. C.E.M. carried out the DFT calculations and A.M.C., C.E.M. and T.N. developed the tight-binding model. All authors contributed to the manuscript.

## Additional information

**Competing interests:** The authors declare no competing interests.

