## [Peer Review File · Nature Communications]

Reviewers' comments:

Reviewer #1 (Remarks to the Author):

In this manuscript, the authors reported the observation of dz^2 orbital along the nodal direction around $\sim 1\text{eV}$ below EF in the single layer cuprate superconductor LSCO. They mainly utilized the polarization selection rule to differentiate the dz^2/xy orbitals from the singlet dx^2-y^2 orbital, and employed photon energy dependence to pin down the k_z dispersion so to differentiate the dz^2 orbital from 'trivial' surface states. With the two-orbital model, they further calculated the dz^2 and dx^2-y^2 component along the high symmetry momenta. The implication of this work is elaborated in two folds: delineate the pseudogap effect from the dz^2 orbital hybridization, and *may* provide insights into the T_c variation in single layer cuprate superconductors.

Overall the work is presented with clarity but for a few minor (technical) concerns. However, my primary concern is the limited new knowledge this work would add to the existing pool. The observation and identification of the dz^2 orbital via XPS, EELS (J. Fink et al, IBM J. Res. Dev. 33, 372) and ARPES (PRL 98, 147001; PRB 75, 174506; R.H. He, PhD Dissertation, Stanford University) have been abundant in literature, even though admittedly largely overlooked by the mainstream due to the (perceived) lack of immediate energetic relevance to the superconductivity. Theory wise, a handful of theoretical calculations ascribed ARPES feature here (-1eV below EF) to the Cu $3dz^2$ orbital (PRB 85, 064501); and the idea of dz^2 hybridization contributing to the family-to-family T_c variance is an old song. The discussion part in this work does little more than echoing/collecting statements (of the expected presence of the dz^2 band) and speculations (antagonist to T_c) from literature.

That being said, the k_z dependence and the polarization analysis in this work made the identification of the dz^2 band more convincing, and it's always good to have nice clean data to back up old statements. One interesting aspect is the larger in-plane dispersion (comparing to that along k_z) and its implication on the real space orbital overlap landscape. But I don't think this lives up to the mission of Nat Commun.

In the following I list a few imperfections that would need further clarification/improvement.

1. The title is highly suspiciously overselling. Observation of dz^2 orbital certainly does not equate 'superconductivity restrained by dz^2 hybridization'. And the hand-wavy comparison with Eu-LSCO and Eu-Sr₂NiO₄ is not helping either. Smaller t' value is long known in LSCO, but for single layer compounds with larger t' (Bi2201), the max T_c remains the same to LSCO. Such a title is likely going to mislead the readers into anticipating some doping or temperature dependence that would *actually* tune the superconductivity.
2. It is not clearly explained why the admixture of dz^2 into dx^2-y^2 would bring the band closer to the EF thus lifting the VHS. There's no 'hybridization gap' to begin with - it's just more of a mixing of band character.
3. It's not immediately obvious how the 'two-band' TB model parameters 'automatically' includes the oxygen ligand contribution. Perhaps add a reference (PRB 85, 064501?) where this statement is made?
4. It is inferred in the manuscript that dz^2 overlaps more with the oxygen p_x/y than with p_z orbital. But isn't this contradicting the root assumption in the two band fitting model where the Wannier functions from dz^2 & p_z , dx^2-y^2 & p_x/y are inherently assumed?
5. As an experiment paper, sample measurement temperature is not mentioned at all (only cleaving temperature) in either the main text or the supplement. This is particularly bothering given the abundance of complex phases that dwells around the SC dome in LSCO.
6. To facilitate the comparison in Fig3, it's more helpful to plot the k_z in/out-of-plane dispersions in an earthly 2D plot so that they are practically, quantitatively, scientifically comparable.

7. The core data of this work are really just two polarization dependent and one a set of photon-energy dependent nodal spectra. Many components in the three main figures already feel like fillers. The calculation is consistent, but it largely falls into a subset of a collection of existing published (calculation) results.

Reviewer #2 (Remarks to the Author):

Matt et al. performed soft x-ray angle-resolved photoemission spectroscopy on $\text{La}_{2-x}\text{Sr}_x\text{CuO}_4$. Via the photoemission selection rules, the authors were able to identify the $d_{x^2-y^2}$ and d_{z^2} bands. Comparing the experimental band structure with a tight-binding model as well as density functional theory, the authors were able to quantify the orbital hybridization, which has been theoretically predicted to suppress superconducting pairing in certain cuprates.

The paper is written very clearly. The data is well presented and the analysis is reasonable. I find that the quantification of orbital hybridization very important for understanding a lot of the peculiar physics in single-layer cuprates. I suggest publication of this manuscript after addressing some issues below.

- 1) Can the authors explain why the d_{z^2} band has not been identified before, despite the tremendous literature on ARPES studies on cuprates?
- 2) It is a little surprising that using rather conventional methods such as density functional theory and tight-binding model, the authors are able to explain most of the ARPES data on cuprates, which are known to deviate from the single-particle picture. Does the theoretical framework only work for overdoped LSCO which is closer to being a Fermi liquid? How well does it work for underdoped LSCO?
- 3) The main comparison is done between theory and overdoped LSCO. Do the tight-binding parameters, especially the hybridization term, change significantly for optimally or under-doped LSCO? That seems to be a more direct test of how the orbital hybridization affects superconductivity and pseudogap.
- 4) Although the overall agreement between data and tight-binding model is decent in Fig. 2, I fail to see a spectral suppression along Γ -X or along Z-R, where a hybridization gap should exist. Can the authors explain why?

Reviewer #3 (Remarks to the Author):

It is very fundamental question to ask which model is the most proper one to describe the low energy electronic structure of the high T_c cuprates. In the earlier works up so-far, single band model is the most supported scenario to describe low energy physics in cuprates. The authors in this work successfully apply the soft X-ray angle-resolved photoemission (ARPES) technique on several $\text{La}_{2-x}\text{Sr}_x\text{CuO}_4$ systems and clearly revealed the existing of band structure around 1 eV below chemical potential with mainly d_{z^2} character by carefully designed their experimental configuration with specific photon polarization. The authors also estimate roughly about 200 meV maxima hybridization amplitude between $d_{x^2-y^2}$ and d_{z^2} orbitals by showing rough qualitative agreement between their two-orbital tight-binding model (TBM) with the measured ARPES results. This non-zero hybridization factor suggest the single band model is not sufficient to describe the low energy physics in La-based cuprates systems and it further explain the suppression of the $T_c(\text{max})$ in La-based cuprates with respect to the other single layer cuprates families. However, there are still several issues worthy to be

brought out and discussed before I totally agree to the acceptance of this work:

1) Most data the authors shown are from soft X-ray measurements, which suggest 50-100 meV or even worse energy resolution. This energy resolution already reaches same order of magnitude of the tiny hybridization factor they finally achieve from the so-called "band structure fit". It would be necessary for the authors to compare their TBM calculation results with measurement result with better energy resolution if possible to have more accuracy in the estimation.

2) From fig 2b and 2d in the main text, and figS3 c1 and c2 in the supplementary, the agreement between two-orbital TMB and the ARPES data in the nodal direction (Γ -M, Z-A) is much better at anti-nodal region. In their data, there are always finite spectral weight existing around chemical potential with water-fall like feature following down to high binding energy at the X- or R- points region. But their TBM calculation always appear well above the chemical potential. The authors better explain why the more hybridization between two eg orbitals, the less agreement between their model with the experimental data.

3) It was well known that oxygen, both plan oxygen and the apical oxygen, plays important role in the low-energy physics of cuprates. Here in this work, authors only very slightly mention, as quote "... all hopping parameters effectively including the oxygen orbitals...". It would be necessary for the authors to put more words on how they include the oxygen orbital effectively in their model, for the reader to better understand their model.

The multi-orbital nature of the low energy band structure indicated by this work is important for people to understand the low energy physics in La-based cuprates, and it certainly shed more light in the field. If the authors can address more on the issues mentioned above, I will fully suggest the acceptance of this work.

Point by point response to the referees' comments for the manuscript:

Reviewer #1 (Remarks to the Author):

In this manuscript, the authors reported the observation of d_{z^2} orbital along the nodal direction around $\sim 1\text{eV}$ below EF in the single layer cuprate superconductor LSCO. They mainly utilized the polarization selection rule to differentiate the d_{z^2}/xy orbitals from the singlet $d_{x^2-y^2}$ orbital, and employed photon energy dependence to pin down the k_z dispersion so to differentiate the d_{z^2} orbital from 'trivial' surface states. With the two-orbital model, they further calculated the d_{z^2} and $d_{x^2-y^2}$ component along the high symmetry momenta. The implication of this work is elaborated in two folds: delineate the pseudogap effect from the d_{z^2} orbital hybridization, and *may* provide insights into the T_c variation in single layer cuprate superconductors.

Authors: We thank referee 1 for his/her comments to our manuscript. Although we do mention the pseudogap, we are not making any strong conclusions on the pseudogap phase. Based on our and literature data, we state that the pseudogap seems unaffected by the presence of orbital hybridization. This is not equivalent to the referee's summary on this point. We do suggest that orbital hybridization have direct consequences for the Fermi surface topology and hence also superconductivity. This has a firm theoretical support. Referee 1's summary is less precise than that given by referee 3.

Overall the work is presented with clarity but for a few minor (technical) concerns.

Authors: We address the minor technical suggestions below.

However, my primary concern is the limited new knowledge this work would add to the existing pool. The observation and identification of the d_{z^2} orbital via XPS, EELS (J. Fink et al, IBM J. Res. Dev. 33, 372) and ARPES (PRL 98, 147001; PRB 75, 174506; R.H. He, PhD Dissertation, Stanford University) have been abundant in literature, even though admittedly largely overlooked by the mainstream due to the (perceived) lack of immediate energetic relevance to the superconductivity.

Authors:

The referee challenges the novelty of our work and mentions four publications that cover "The observation and identification of the d_{z^2} orbital and its hybridization with the $d_{x^2-y^2}$ band" largely overlooked by the mainstream.

Before discussing these four papers, we would like to point out that any d^9 system has to have a (partially) filled d_{z^2} orbital. The existence of occupied d_{z^2} states is therefore not of big surprise and has been discussed before. Yet, the d_{z^2} band has not been directly reported by ARPES experiments.

We agree with the referees' statement that the d_{z^2} states have received little attention by the mainstream community due to a perceived lack of immediate energetic relevance to superconductivity. This view has exactly been widespread because previous experimental observations of the d_{z^2} states have not been able to reveal the complete d_{z^2} band dispersion. For this

reason, the inter-orbital hopping between d_{z^2} and $d_{x^2-y^2}$ states was entirely unknown. Our study provides the full three-dimensional d_{z^2} band structure and reveals directly the inter-orbital hopping between d_{z^2} and $d_{x^2-y^2}$ states. It is approximately 200 meV in overdoped LSCO.

The comments of the referee do suggest that he/she agrees that direct experimental demonstration of the energetic relevance of the d_{z^2} band for superconductivity is important. The impact of our experimental work is that it will change the mainstream view on this problem. Let us now comment on the four mentioned publications.

J. Fink et al, IBM J. Res. Dev. 33, 372:

This publication from 1989 indeed reports XPS and EELS experiments in several cuprate compounds. Identification of the oxygen k-edge is made. However, at this early stage, the data quality was not yet sufficient to resolve the apical and planar resonances. This is why, in our manuscript we discuss the subsequent XAS work (PRL **68**, 2543 (1992)) in which apical and planar resonances are clearly resolved and studied with light polarization analysis. We, by no means, agree that the early XPS & EELS experiments are comparable to the direct ARPES measurements of the d_{z^2} band. Modern RIXS studies of the dd -excitations are in our opinion more revealing and we have in the revised manuscript included references to such experiments.

W. Meevasana et al., PRB 75, 174506 and B. P Xie et al., PRL 98, 147001 (2007)

None of these two papers mention the d_{z^2} orbital with a single word. It is true that both a A and B-band are reported in these papers. The A-band obviously originates from the $d_{x^2-y^2}$ orbital while the orbital origin of the B-band is not identified or discussed in these papers. We assume that referee 1, him/her-self assigns d_{z^2} orbital character to the B-band.

There are, however, several observations that challenge the identification of the B band with d_{z^2} orbital character:

1. In PRL 98, 147001 (2007), it is stated that the A and B bands are hybridizing along the nodal direction. No hybridization between $d_{x^2-y^2}$ and d_{z^2} orbitals are expected along the nodal direction.
2. In PRB 75, 174506 (2007), nodal k_z -dependence has been carried out with photon-energies 40-55eV. The authors observe no k_z -dispersion of the B band which further complicates its identification with d_{z^2} character.
3. Furthermore, band structure calculations are presented in both papers. However, the orbital projections are completely omitted. Even if included, interpretation of the Bi2201 data would not be straightforward. The DFT calculation [presented in PRB 75, 174506 (2007)] predicts at least two quasi-degenerate bands consistent with the observed B-band.

The papers therefore give no clear proof that the B-band could be connected to the d_{z^2} orbital. In the revised manuscript, we mention the waterfall feature and cite these two papers.

R. H. He's PhD thesis.

In appendix A of this PhD thesis, experimental matrix element analysis of primarily LSCO $x=1/8$ is presented. None of the ARPES data extend beyond binding energies of 800 meV. Obviously, the d_{z^2} band is therefore not reported directly. R. H. He did have the idea that d_{z^2} orbital character is hybridized into the Fermi surface spectral weight. By a photon polarization dependent analysis, R.H. He disentangles the contribution of the d_{z^2} and the $d_{x^2-y^2}$ orbital at the Fermi level. His analysis relies entirely on matrix element effects and reveals that the band of the d_{z^2} orbital is distinct from the one of the $d_{x^2-y^2}$ orbital. The Fermi surface formed by the d_{z^2} band is electron like even in the very underdoped regime and shows only very little doping dependence. R.H. He writes himself that this is

“surprising”. Such a result is rather puzzling and in disagreement with our observation of a single band Fermi surface.

In summary, none of the four mentioned publications reports on the full dispersion of the d_{z^2} band and its contribution to the low energy electronic structure (hybridization effect). It is therefore incorrect to state that our observations appear abundantly in the literature.

Theory wise, a handful of theoretical calculations ascribed ARPES feature here (-1eV below EF) to the Cu $3d_{z^2}$ orbital (PRB 85, 064501); and the idea of d_{z^2} hybridization contributing to the family-to-family T_c variance is an old song. The discussion part in this work does little more than echoing/collecting statements (of the expected presence of the d_{z^2} band) and speculations (antagonist to T_c) from literature.

Authors: Whether or not a theoretical idea is considered to be an old or new song, it has to be tested by direct experiments. As long as the d_{z^2} band is not directly observed by ARPES, the idea remains hypothetical. With our results, it is now established that the d_{z^2} band is found fairly close to the Fermi level and hybridizes with the $d_{x^2-y^2}$ band. It should hence be included in any realistic model of LSCO. The single band picture might still be a good approximation for other cuprate systems.

That being said, the k_z dependence and the polarization analysis in this work made the identification of the d_{z^2} band more convincing, and it's always good to have nice clean data to back up old statements.

Authors: We assume that the referee 1 with “old statements” refers to the aforementioned four experimental publications. We stress again, that none of these publications reported the d_{z^2} band. In particular, the PRL 98, 147001 and PRB 75, 174506 papers have absolutely no mentioning of the d_{z^2} orbital.

One interesting aspect is the larger in-plane dispersion (comparing to that along k_z) and its implication on the real space orbital overlap landscape. But I don't think this lives up to the mission of Nat Commun.

Authors: We are pleased that the referee 1 appreciates certain aspects of our manuscript. At the same time, we disagree strongly with the view that this point is the only novelty of our work.

In the following I list a few imperfections that would need further clarification/improvement.

1a. The title is highly suspiciously overselling. Observation of d_{z^2} orbital certainly does not equate 'superconductivity restrained by d_{z^2} hybridization'.

1b. And the hand-wavy comparison with Eu-LSCO and Eu-Sr₂NiO₄ is not helping either.

1c. Smaller t' value is long known in LSCO, but for single layer compounds with larger t' (Bi2201), the max T_c remains the same to LSCO.

1d. Such a title is likely going to mislead the readers into anticipating some doping or temperature dependence that would *actually* tune the superconductivity.

Authors: Our title refers to the theoretical fact that hybridization restrains superconductivity. To avoid the association with the doping phase diagram, we have changed to title to: “Direct observation of orbital hybridisation in a cuprate superconductor”

2. It is not clearly explained why the admixture of d_{z^2} into $d_{x^2-y^2}$ would bring the band closer to the EF thus lifting the VHS. There's no 'hybridization gap' to begin with - it's just more of a mixing of band character.

Authors: The reviewer has two comments related: (1) the physical mechanism by which inter-orbital hopping brings the van Hove singularity closer to the Fermi energy and (2) whether we use the correct terminology by using 'hybridization'.

To address (1), we consider the hypothetical situation where the inter-orbital hopping is switched off. In our tight-binding model, the two bands then have well defined orbital character (d_{z^2} and $d_{x^2-y^2}$). At the anti-nodal X point where the van Hove singularity is located both bands are found below the Fermi level. Once the inter-orbital hopping is included (which is largest at the X point), it pushes the two bands further apart, as in a level repulsion.

As a result, the band closest the Fermi level moves up and eventually across E_F . The lower band by contrast moves to deeper binding energy. We illustrate this here by plotting the energy of the two bands at the X point as a function of the inter-orbital hopping strength for LASC0 (using the tight-binding model with the other parameters as given in the manuscript).

To address (2), we emphasize that we never use the terminology 'hybridization gap' in the manuscript. However, we refer to the band 'hybridization' as a result of the inter-orbital hopping. There is no fundamental physical distinction between a hybridization of states and a mixing of them in our context. One starts with a system that has a degeneracy (here: the crossing point of a d_{z^2} and $d_{x^2-y^2}$ band) protected by some quantum number (here: the orbital character of the degenerate bands). Then a term that does not conserve this quantum number is introduced (here: the inter-orbital hopping) and the degeneracy is lifted. One says that the two initially degenerate states are hybridized by the perturbation, which also leads to a mixed character of the new eigenstates. This effect is best appreciated by looking at the red/black bands in Fig. 2e of our manuscript along the Γ -X line.

3. It's not immediately obvious how the 'two-band' TB model parameters 'automatically' includes the oxygen ligand contribution. Perhaps add a reference (PRB 85, 064501?) where this statement is made?

Authors: Both referee 1 and 3 suggest that our discussion of the oxygen in the hopping processes was too succinct. We agree with this comment and have expanded the text on this point by mentioning relevance of the super-exchange mechanism. The passage now reads:

"This is due to the fact that the inter-plane hopping is mostly mediated by hopping between apical oxygen p_z orbitals, which in turn only hybridize with the d_{z^2} orbitals, not with the $d_{x^2-y^2}$ orbitals. Such

an argument highlights that the tight-binding model is not written in atomic orbital degrees of freedom, but in Wannier orbitals, which are formed from the Cu d-orbitals and the ligand oxygen p orbitals. As follows from symmetry considerations and is discussed in Ref. [PRB **85**, 064501], the Cu d_{z^2} orbital together with the apical oxygen p_z orbital forms a Wannier orbital with d_{z^2} symmetry, while the Cu $d_{x^2-y^2}$ orbital together with the four neighboring p_{σ} orbitals of the in-plane oxygen forms a Wannier orbital with $d_{x^2-y^2}$ symmetry. One should thus think of this tight-binding model as written in terms of these Wannier orbitals, thus implicitly containing superexchange hoppings via the ligand oxygen p orbitals.”

4. It is inferred in the manuscript that dz^2 overlaps more with the oxygen px/y than with pz orbital. But isn't this contradicting the root assumption in the two band fitting model where the Wannier functions from dz^2 & pz , dx^2-y^2 & px/y are inherently assumed?

We wrote:

“Interestingly, this dz^2 -derived band has stronger in-plane than out-of-plane dispersion, suggesting that the dz^2 -orbital hybridizes more with in-plane px and py oxygen orbitals than with the apical oxygen pz orbital.”

This statement is as the referee point-out not quite accurate. To correct, we have reformulated such that it reads:

“Interestingly, this dz^2 -derived band has stronger in-plane than out-of-plane dispersion, suggesting that there is a significant hopping to in-plane p_x and p_y oxygen orbitals.”

5. As an experiment paper, sample measurement temperature is not mentioned at all (only cleaving temperature) in either the main text or the supplement. This is particularly bothering given the abundance of complex phases that dwells around the SC dome in LSCO.

Authors: Cleaving and measurement temperature is identical. This is now stated in the method section.

6. To facilitate the comparison in Fig3, it's more helpful to plot the k_z in/out-of-plane dispersions in an earthly 2D plot so that they are practically, quantitatively, scientifically comparable.

Authors: We have now included the requested 2D re-plotting in the supplementary information and reference to it has been made in the figure caption.

7. The core data of this work are really just two polarization dependent and one a set of photon-energy dependent nodal spectra. Many components in the three main figures already feel like fillers. The calculation is consistent, but it largely falls into a subset of a collection of existing published (calculation) results.

Authors: The referee claim / states that our work consists of just two polarization dependent nodal spectra in addition to nodal k_z dependence. This statement is incorrect. Fig. 2 for example presents mapping of the d_{z^2} band along three in-plane high-symmetry directions. Each of these planes are presented for two different high-symmetry values of $k_z=0$ and π . Our presentation is thus not limited to nodal spectra.

We have interpreted the referees statement as a suggestion to include more data taken along high-symmetry directions different from the nodal line. Based also on comments from referee 2 and 3, we have included a new Fig. 3 that clearly demonstrates the repulsive coupling between the dz^2 and dx^2-y^2 bands.

Reviewer #2 (Remarks to the Author):

Matt et al. performed soft x-ray angle-resolved photoemission spectroscopy on $\text{La}_{2-x}\text{Sr}_x\text{CuO}_4$. Via the photoemission selection rules, the authors were able to identify the $d_{x^2-y^2}$ and d_{z^2} bands. Comparing the experimental band structure with a tight-binding model as well as density functional theory, the authors were able to quantify the orbital hybridization, which has been theoretically predicted to suppress superconducting pairing in certain cuprates.

The paper is written very clearly. The data is well presented and the analysis is reasonable. I find that the quantification of orbital hybridization very important for understanding a lot of the peculiar physics in single-layer cuprates. I suggest publication of this manuscript after addressing some issues below.

Authors: We thank the referee for his/her recommendation and respond below to the suggestions.

1) Can the authors explain why the d_{z^2} band has not been identified before, despite the tremendous literature on ARPES studies on cuprates?

Authors: This is a good question that we have addressed in the revised manuscript. In the field of cuprate superconductivity there has been an enormous focus on low-energy spectral features. The superconducting gap, the pseudogap and various kink energy scales have been studied in greatest detail. In the quest to resolve these energy scales, there has been a drive to optimize the energy resolution. For synchrotron ARPES, lower photon-energy provides higher energy resolution. For this reason, a large majority of experiments have been carried out using photon energies less than 100 eV.

In the period 2007- 2010 there was a vivid discussion of the nodal waterfall feature found in virtually all known cuprates. This led many ARPES groups to explore the photoemission spectra down to 1eV. In some cases, as pointed out by referee 1, spectra were collected down to 2 or 2.5 eV -- exactly the energy range probed for our experiments.

For some reason the binding energy range 1-2.5 eV was not covered for LSCO. Moreover, following the tradition of using photon energies below 100 eV was followed even high-energy resolution is not strictly required. It is not impossible to detect the d_{z^2} band with 55 eV photons known to give high matrix element for LSCO in the second Brillouin zone. In fact, this was the photon energies of our first experiments. Driven by the questions of bulk sensitivity and k_z dispersion, we decided to investigate then band structure with photon energies larger than 100 eV. In this fashion, we discovered that both the $d_{x^2-y^2}$ and d_{z^2} band appears very clearly when 160 eV photons are used. Moreover, we also found that these two bands can be measured for a wide energy range in the soft-x-ray regime of 400 – 600 eV. We stress that soft x-ray ARPES has only rarely been applied to the cuprate problem. Notice that this instrumentation is not widespread. We are only aware of two of such instruments (ADDRESS @ PSI and BL17SU @ SPring-8, Japan).

2) It is a little surprising that using rather conventional methods such as density functional theory and tight-binding model, the authors are able to explain most of the ARPES data on cuprates, which are known to deviate from the single-particle picture. Does the theoretical framework only work for overdoped LSCO which is closer to being a Fermi liquid? How well does it work for underdoped LSCO?

Authors: We have only studied one underdoped compound so far (LSCO $x=0.12$), see supplementary information Fig S1. Obviously, for energies between E_F and -0.5 eV there is a strong variation of the ARPES spectra as a function of doping. The global band structure including the d_{z^2} band, however, appears very similar to data recorded on the overdoped side. It thus suggests that for high binding energies (below ~ 0.5 eV), DFT calculations are actually working very well for LSCO irrespectively of doping.

3) The main comparison is done between theory and overdoped LSCO. Do the tight-binding parameters, especially the hybridization term, change significantly for optimally or under-doped LSCO? That seems to be a more direct test of how the orbital hybridization affects superconductivity and pseudogap.

Authors: This is a good point by the referee. As it stands, we have collected large and complete data sets for overdoped LSCO. A few spectra along selected high symmetry directions for a few photon energies has been measured on an underdoped sample. We therefore do not yet have enough doping dependent data to make any strong conclusion in this direction. For future experiments, we are planning to study systematically the doping dependence.

4) Although the overall agreement between data and tight-binding model is decent in Fig. 2, I fail to see a spectral suppression along Gamma-X or along Z-R, where a hybridization gap should exist. Can the authors explain why?

Authors: This is a correct observation. The maxima hybridization amplitude is in the order of 200 meV. This is comparable to the width of the spectral weight of the d_{z^2} band. We thus believe that it is the broadness of the spectral band signatures that prevents us from seeing directly the hybridization gap in the soft x-ray regime. In the revised version of the manuscript, we have included data recorded with 160 eV photons (ultra-violet regime) along the mentioned Z-R direction (new Fig. 3). The bands are clearly separated and hence evidence the hybridization gap.

Reviewer #3 (Remarks to the Author):

It is very fundamental question to ask which model is the most proper one to describe the low energy electronic structure of the high T_c cuprates. In the earlier works up so-far, single band model is the most supported scenario to describe low energy physics in cuprates. The authors in this work successfully apply the soft X-ray angle-resolved photoemission (ARPES) technique on several $\text{La}_{2-x}\text{Sr}_x\text{CuO}_4$ systems and clearly revealed the existing of band structure around 1 eV below chemical potential with mainly d_{z^2} character by carefully designed their experimental configuration with specific photon polarization. The authors also estimate roughly about 200 meV maxima hybridization amplitude between $d_{x^2-y^2}$ and d_{z^2} orbitals by showing rough qualitative agreement between their two-orbital tight-binding model (TBM) with the measured ARPES results. This non-zero hybridization factor suggest the single band model is not sufficient to describe the low energy physics in La-based cuprates systems and it further explain the suppression of the $T_c(\text{max})$ in La-based cuprates with respect to the other single layer cuprates families. However, there are still several issues worthy to be brought out and discussed before I totally agree to the acceptance of this work:

1) Most data the authors shown are from soft X-ray measurements, which suggest 50-100 meV or even worse energy resolution. This energy resolution already reaches same order of magnitude

of the tiny hybridization factor they finally achieve from the so-called “band structure fit”. It would be necessary for the authors to compare their TBM calculation results with measurement result with better energy resolution if possible to have more accuracy in the estimation.

Authors: We have included data (Fig 3) recorded at 160 eV (π polarization) along the anti-nodal direction where the hybridization gap can be directly estimated. At this photon energy, the energy resolution is significantly better than in the soft-xray regime. From these data, the hybridization gap is clearly visible.

2) From fig 2b and 2d in the main text, and figS3 c1 and c2 in the supplementary, the agreement between two-orbital TBM and the ARPES data in the nodal direction (Γ -M, Z-A) is much better at anti-nodal region. In their data, there are always finite spectral weight existing around chemical potential with water-fall like feature following down to high binding energy at the X- or R- points region. But their TBM calculation always appear well above the chemical potential. The authors better explain why the more hybridization between two eg orbitals, the less agreement between their model with the experimental data.

Authors: The referee’s observation is correct. For the analysis of tight-binding parametrization, we faced a dilemma. It is not possible with simple TBM to capture both the effective band dispersion near the Fermi level and at the same time the band structure of the d_{z^2} -orbital. For example, if we use the nodal Fermi velocity to define band width of the $d_{x^2-y^2}$ band, a huge discrepancy will be found around and at the observed band bottom shown in Fig. 2f. Therefore, we decided to set the focus more on the high-energy part of the spectrum and hence opened for larger discrepancy at lower energies. Our TBM does in this fashion not take into account self-energy effects. It is possible that the self-energy is momentum dependent and hence generates a larger discrepancy in the antinodal region.

3) It was well known that oxygen, both plan oxygen and the apical oxygen, plays important role in the low-energy physics of cuprates. Here in this work, authors only very slightly mention, as quote “... all hopping parameters effectively including the oxygen orbitals...”. It would be necessary for the authors to put more words on how they include the oxygen orbital effectively in their model, for the reader to better understand their model.

Authors: As similar suggestion was made by referee 1. We agree that the discussion of oxygen hoppings was too succinct. Therefore, we have expanded the text on this point by mentioning relevance of the super-exchange mechanism. The passage now reads:

“This is due to the fact that the inter-plane hopping is mostly mediated by hopping between apical oxygen p_z orbitals, which in turn only hybridize with the d_{z^2} orbitals, not with the $d_{x^2-y^2}$ orbitals. Such an argument highlights that the tight-binding model is not written in atomic orbital degrees of freedom, but in Wannier orbitals, which are formed from the Cu d orbitals and the ligand oxygen p orbitals. As follows from symmetry considerations and is discussed in Ref. [PRB **85**, 064501], the Cu d_{z^2} orbital together with the apical oxygen p_z orbital forms a Wannier orbital with d_{z^2} symmetry, while the Cu $d_{x^2-y^2}$ orbital together with the four neighboring p_o orbitals of the in-plane oxygen forms a Wannier orbital with $d_{x^2-y^2}$ symmetry. One should thus think of this tight-binding model as written in terms of these Wannier orbitals, thus implicitly containing superexchange hopping via the ligand oxygen p orbitals”

The multi-orbital nature of the low energy band structure indicated by this work is important for people to understand the low energy physics in La-based cuprates, and it certainly shed more light in the field. If the authors can address more on the issues mentioned above, I will fully suggest the acceptance of this work.

Authors: We thank the referee for his / her constructive suggestions that were helpful to improve our manuscript.

REVIEWERS' COMMENTS:

Reviewer #1 (Remarks to the Author):

I agree with the author's response for most parts. While I offered a different voice, at this point, if the majority (of the referees) consider this work ('direct identification of d_{z^2} orbital') to be of novelty, I shall hold no more objections. My only remaining comment would be that at least the authors now give literature enough credit for covering this issue. Otherwise, I have no issue with the data, given the newly tuned-down narrative especially the title.

Reviewer #2 (Remarks to the Author):

Matt et al. has addressed all my questions. Systematically characterizing the d_{z^2} orbital and its hybridization with the dx^2-y^2 orbital is very important for the study of cuprate superconductors. I think the current work is clearly novel compared to the previous reports which may hint at seeing some contributions from the d_{z^2} orbital but not systematically characterizing the dispersions. I fully support the publication of this work in Nature Communications.

Reviewer #3 (Remarks to the Author):

The authors had given a more detail explanation to the model they used in their work and also expressed well the reason why they used the soft X-ray ARPES for their studies. After carefully reading the authors's answer and explanation to all three referee reports, I think the authors had given thorough explainatin to all my concerns and provided proper modification to the manuscript. I have no more further commend and approve the acceptance of this work in Nature communication.

Point by point response to the referees' comments for the manuscript:

Reviewer #1: I agree with the author's response for most parts. While I offered a different voice, at this point, if the majority (of the referees) consider this work ('direct identification of d_{z^2} orbital') to be of novelty, I shall hold no more objections. My only remaining comment would be that at least the authors now give literature enough credit for covering this issue. Otherwise, I have no issue with the data, given the newly tuned-down narrative especially the title.

Authors: We are pleased to know that referee 1 agrees with our responses and now recommends our manuscript for publication. As remaining comment, Referee 1 suggests us to give sufficient credit to literature. Following the referee's suggestion, we have now included a reference to J. Fink et al, IBM J. Res. Dev. 33, 372. In this fashion, we are citing all the published work mentioned in the reports of referee 1.

Reviewer #2: Matt et al. has addressed all my questions. Systematically characterizing the dz_2 orbital and its hybridization with the dx_2-y_2 orbital is very important for the study of cuprate superconductors. I think the current work is clearly novel compared to the previous reports which may hint at seeing some contributions from the dz_2 orbital but not systematically characterizing the dispersions. I fully support the publication of this work in Nature Communications.

Reviewer #3: The authors had given a more detail explanation to the model they used in their work and also expressed well the reason why they used the soft X-ray ARPES for their studies. After carefully reading the authors's answer and explanation to all three referee reports, I think the authors had given thorough explainatin to all my concerns and provided proper modification to the manuscript. I have no more further commend and approve the acceptance of this work in Nature communication.

Authors: We thanks both referee 2 and 3 for their recommendation of our manuscript.